# Enhancing the Antioxidant, Antibacterial, and Wound Healing Effects of *Melaleuca alternifolia* Oil by Microencapsulating It in Chitosan-Sodium Alginate Microspheres

**DOI:** 10.3390/nu15061319

**Published:** 2023-03-07

**Authors:** Anbazhagan Sathiyaseelan, Xin Zhang, Myeong-Hyeon Wang

**Affiliations:** Department of Bio-Health Convergence, Kangwon National University, Chuncheon 200-701, Republic of Korea

**Keywords:** chitosan, sodium alginate, tea tree oil, antibacterial, microsphere, wound healing

## Abstract

In this study, antibacterial and antioxidant molecules-rich *Melaleuca alternifolia* oil (tea tree oil (TTO)) loaded chitosan (CS) based nanoemulsions (NEMs) were prepared and encapsulated by sodium alginate (SA) microsphere for antibacterial wound dressing. CS-TTO NEMs were prepared by oil-in-water emulsion technique, and the nanoparticle tracking analysis (NTA) confirmed that the CS-TTO NEMs had an average particle size of 89.5 nm. Further, the SA-CS-TTO microsphere was confirmed through SEM analysis with an average particle size of 0.76 ± 0.10 µm. The existence of TTO in CS NEMs and SA encapsulation was evidenced through FTIR analysis. The XRD spectrum proved the load of TTO and SA encapsulation with CS significantly decreased the crystalline properties of the CS-TTO and SA-CS-TTO microsphere. The stability of TTO was increased by the copolymer complex, as confirmed through thermal gravimetric analysis (TGA). Furthermore, TTO was released from the CS–SA complex in a sustained manner and significantly inhibited the bacterial pathogens observed under confocal laser scanning microscopy (CLSM). In addition, CS-TTO (100 µg/mL) showed antioxidant potential (>80%), thereby increasing the DPPH and ABTS free radicals scavenging ability of SA-CS-TTO microspheres. Moreover, CS and SA-CS-TTO microsphere exhibited negligible cytotoxicity and augmented the NIH3T3 cell proliferation confirmed in the in vitro scratch assay. This study concluded that the SA-CS-TTO microsphere could be an antibacterial and antioxidant wound dressing.

## 1. Introduction

Skin regulates multiple functions including maintaining the temperature and protecting internal organs, and it protects from damage/trauma through the self-healing ability of its complex cellular structure (epidermis and dermis) [1]. Wound healing is a complex biological process which is rapidly established the skin structure by the system after skin injury to prevent internal organs from harmful environmental and biological factors [2]. In addition, the wound healing process is categorized into four major phases (Haemostasis, inflammation, proliferation, and remodeling) [3]. Moreover, important molecules such as coagulation factors, cytokines, pro-inflammatory and anti-inflammatory factors, growth factors, and extracellular matrix initiators are involved in each cellular process [4]. However, metabolic conditions (diabetes, vascular diseases, pulmonary disease, immune-compromised or autoimmune diseases) and microbial infections interrupt the normal healing process and may cause severe damage to the internal organs when residing in the systems [5]. The prolonged microbial infections caused biofilm formation through the colonization of microorganisms, which is very hard to eradicate from the body and leads to severe internal damage. The bacterial infection on the wound further caused severe infections such as osteomyelitis, cellulitis, and sepsis [6,7]. Several antibiotics are recommended for treating bacterial infections in the early stages of wound healing with anti-inflammatory agents, but the emergence of antibacterial resistance destroys their therapeutic functions and delays the healing process. Hence, biocompatible natural antibacterial hemostatic materials are required for advanced wound dressing [8].

Essential oils (EO) contain higher contents of lipophilic and aromatic bioactive molecules (polyphenols, monoterpenes, triterpenes, sesquiterpenes, aldehyde, and ketones), extracted from various plants (thyme, cloves, lemongrass, oregano, cinnamon, black pepper, tea tree, eucalyptus, rosemary, ginger, etc.) [9,10]. Moreover, EO is considered a natural multifunctional material due to the reported biomedical (antimicrobial, anti-inflammatory, anti-septic), industrial applications (flavor), and agricultural applications (Insecticide, pesticide, and antifungal) [11,12]. Particularly, tea tree oil (TTO) extracted from *Melaleuca alternifolia* belongs to the Myrtaceae family, which is potent in antimicrobial skin infection, antioxidant, and anti-inflammatory properties due to its combination of various volatile compounds (terpinen-4-ol, γ-terpinene, α-terpinene, 1,8-cineole, Terpinolene, p-cymene, α-pinene, α-Terpineol, Limonene, and other trace molecules) [13,14]. However, the poor physical properties (hydrophobicity, volatility, and lower degradations) and biological effects (severe toxicity (skin allergies, ototoxicity, developmental toxicity, cytotoxicity, mutagenicity), and ecotoxicity when administered in higher doses) limit their biomedical application (antimicrobial, anti-cancer, and anti-inflammatory) as well as other uses [15]. On the other hand, these drawbacks can be resolved by encapsulating the EO in a polymeric nanomaterial.

Nanoemulsions (NEMs) are widely used to deliver lipophilic bioactive molecules to therapeutic targets with improved degradation and sustained release, which minimizes the various toxicity while improving the therapeutic efficacy [16]. In addition, essential oil-based NEMs showed advanced properties such as antimicrobial, antioxidant, food preservation, and biocompatibility. In general, NEMs are stabilized by various chemical surfactants/additives (Tween (20, 40, and 80), glutaraldehyde, epichlorohydrin, glycerol, sodium dodecyl sulfate, etc.) alone or together [17,18,19]. However, the higher concentration of surfactant and inappropriate combination of other materials may cause toxicity to a living system [16]. To improve biocompatibility and non-toxicity, the bioinspired surfactant is used as the co-surfactant, thereby decreasing the use of synthetic surfactants [20,21]. Recent studies are evidenced that biopolymer-based NEMs have considerable attention due to their controlled bioactive molecules delivery and enchaining therapeutic efficacy [19]. Particularly, chitosan, a cationic polymer derived from chitin through the deacetylation process, is highly found in crustaceans and fungi [22,23]. Studies reported that chitosan considerably improved the stabilization properties of EO in the water, along with other surfactants [14,24]. Considering the deep wound dressing, the biopolymer-based microsphere is highly effective in terms of cellular regrowth, function, and improved physical strength. This study aimed to prepare the SA encapsulated TTO loaded CS microsphere for considering deep wound healing and antibacterial efficacy.

## 2. Materials and Methods

### 2.1. Materials

Chitosan (Low molecular weight 50–190 kDa; degree of deacetylation 75–85%), sodium alginate, glycerol, and Tween 80 were purchased from Sigma-Aldrich (St. Louis, MO, USA). Tea tree oil was purchased from Grasse International, Chennai, Tamil Nadu, India. Ethanol and calcium chloride (CaCl_2_) were obtained from Samchun pure chemical Co., Ltd., Seoul, Republic of Korea. Acetic acid was obtained from Daejung Chemical & Metals Co., Ltd., Siheung-si, Republic of Korea. CellomaxTM (cell viability assay kit) was purchased from MediFab, Seoul, Republic of Korea. Fluorescent stains (rhodamine 123, propidium iodide, dichlorofluorescein diacetate, and acridine orange) were purchased from Sigma-Aldrich (St. Louis, MO, USA). Moreover, the average molecular weight of chitosan was determined at 72.43 kDa and reported in our earlier study [25].

### 2.2. Preparation of TTO-Loaded CS NEMs

TTO-loaded CS NEMs were prepared by the oil-in-water emulsification method. For the aqueous phase, CS (2 mg/mL) was dissolved in acetic acid (0.5%) containing an aqueous medium overnight. Then, undissolved particles were filtered using a syringe filter (0.45 µm) and mixed with 0.5% of CaCl_2_, 0.3% of Tween 80, and 0.3% of glycerol under stirring for 30 min. TTO (1%) was dissolved in ethanol for the oil phase. For the NEMs preparation, the oil phase was slowly added to the aqueous phase and stirred by a high-speed stirrer for 30 min. Then, the NEMs was sonicated by an ultrasonicator (Sonic and materials, US/VCX-130, Newtown, CT, USA) at 80% amplitude for 10 min with a 1 min turn-off cycle. The obtained NEMs was kept at room temperature for further use.

### 2.3. Preparation of SA-CS-TTO Microsphere

The CS-TTO encapsulated sodium alginate microsphere was prepared by the thin film dispersion technique. In brief, 1% of sodium alginate (SA) was dissolved in preheated distilled water (80 °C) and stirred overnight at room temperature; undissolved particles were removed by filtration using a muslin cloth. For the preparation of the microsphere, 10 mL of CS-TTO NEMs (2 mg/mL) was taken into the 5 mL syringe and dropwise added to the 20 mL of 1% SA under stirring. Subsequently, the developed film on the SA solution was dispersed using ultrasonication for 10 min. Further, dispersed microspheres were collected by centrifugation at 8000 rpm for 15 min. Then, the microspheres were thoroughly washed with distilled water and freeze-dried.

### 2.4. Characterization of Nanoemulsion and Microsphere

The prepared CS-TTO NEMs was characterized using a Nano tracking analyzer (NTA, Malvern (NS300), Malvern, UK). The external morphology of microspheres was observed under a field emission scanning electron microscope (FE-SEM, Hitachi S-4800, Tokyo, Japan). The functional properties of CS, SA, CaCl_2_, TTO, CS-TTO, and CS-TTO-SA were studied through Attenuated Total Reflectance-Fourier transform infrared (FTIR-ATR) spectroscopy (Thermo Scientific (iN10/iS50), Waltham, MA, USA). In addition, crystalline properties of CS, SA, CS-SA, and SA-CS-TTO microsphere were determined in X-ray diffraction analysis (XRD, X’pert-pro MPD-PANalytical, Malvern, UK). Further, the thermal stability of the SA-CS-TTO microsphere was evaluated by a thermal analyzer (TA instruments (DSC Q2000/SDT Q600), New Castle, DE, USA).

### 2.5. Water Adsorption and In Vitro Degradation Properties

The water absorption properties of SA-CS-TTO microspheres were determined according to earlier studies [26]. In brief, 100 mg of dried SA-CS-TTO microspheres were added to the beaker containing the 10 mL of PBS and incubated at 37 °C for 30 min. After incubation, the wet SA-CS-TTO microspheres were collected by centrifugation, and measured their wet weight. The water absorption ability of microspheres was determined according to the earlier method [26]. For the in vitro degradation, the known concentration of microsphere (100 mg) was added to the lysozyme (10 µg/mL)-incorporated PBS (10 mL) and incubated at 37 °C for 7, 14, and 21 days. After each incubation, the microspheres were collected by centrifugation, dried in the freeze drier, and measured their dry weight. The in vitro degradation ability of the CS-TTO-SA microsphere was quantified according to the following Formula (1).
% of in vitro degradation = ((Initial weight − Final weight))/(Initial weight) × 100(1)

### 2.6. TTO Loading, Encapsulation, and In Vitro Release

The TTO loading and encapsulation efficiency (LE and EE) in SA-CS-TTO were determined according to the earlier report [27], with some modifications. In brief, the known concentration of microsphere (10 mg) was immersed into the volumetric flask with a known volume of hexane (10 mL) at 37 °C for 1 day and repeated three times. After incubation, all the solvents were filtered using Whatman No.1 filter paper. Then, the content of TTO in the solvent was measured using a UV-Visible spectrophotometer at the absorbance of 260 nm. The absorption maximum was obtained through scanning (200–700 nm) the TTO dissolved in hexane with various concentrations. The unknown concentration of TTO in the supernatant was determined from the known concentration of TTO (y = 4.3334x + 0.0207; R^2^ = 0.999) (Appendix A). The EE and LE of TTO were calculated according to the above-mentioned report [27]. Further, the TTO release from the SA-CS-TTO microsphere was determined according to the earlier method [27] with some modifications. In brief, the known weight of the SA-CS-TTO microsphere was dispersed in the different buffer solutions (pH 5.4 and 7.4) and incubated for 5 days. At each scheduled time interval, samples were drawn from the bath and collected by centrifugation. To extract the TTO from the samples, both pellet and the supernatant were mixed with a known volume of hexane and incubated for 15 min. Then extracted TTO was quantified by measuring absorption maximum at 260 nm. Furthermore, the cumulative TTO release from the microsphere was determined according to an earlier report [27].

### 2.7. Biological Application of SA-CS-TTO Microsphere

#### 2.7.1. Antibacterial Assay

The bacterial inhibitory properties of different concentrations of TTO, CS, CS-TTO, SA, and SA-CS-TTO were determined in a well diffusion assay. In brief, bacterial pathogens were purchased from the American type culture collection center (ATCC). These bacterial strains *Bacillus cereus* ATCC 14579, *Staphylococcus aureus* ATCC 19095, *Escherichia coli* ATCC 43888, and *Salmonella enterica* ATCC 14028 were initially cultured in pre-sterilized nutrient broth and incubated at 37 °C overnight. After incubation, each bacterial suspension (50 µL) was added to the pre-poured sterile MHA plate and evenly spread all over the plate. The wells were made in the bacterial culture-inoculated MHA medium using a sterile steel borer. Subsequently, the different concentrations of each sample (50 µL) were added to each well and incubated at 37 °C for 12 h. The zone of inhibition was measured using the ruler around each well after predetermined incubation. Moreover, the bacterial growth curve with each sample and without the sample was evaluated by the broth dilution method according to the earlier protocol [28]. Further, bacterial cell viability and the effect of treatment were studied through SYTO 9/PI fluorescent bacterial live and dead assay [29]. In brief, both Gram-positive (*S. aureus*) and Gram-negative (*E. coli*) bacterial pathogens were incubated with each sample for 12 h. After incubation, the bacteria were collected by centrifugation and washed with PBS twice. Subsequently, 3 µL/mL of an equal ratio of SYTO 9 and PI were added to the bacterial culture and incubated for 15 min. Then the staining solution was discarded by washing with PBS and dispersed with PBS. In total, 10 µL of bacterial suspension was added to the confocal disc and observed under a confocal laser scanning microscope in green (λ_ex_/λ_em_ = 488/500 nm) and red fluorescence (λ_ex_/λ_em_ = 550/600 nm).

#### 2.7.2. Antioxidant Assays

The free radical scavenging ability of TTO, CS, SA, CS-TTO, and SA-CS-TTO was determined by DPPH and ABTS free radical scavenging assay according to the earlier report [14,30]. ABTS radicals were prepared according to the earlier method [14]. In brief, different concentrations of each sample were added to 100 µL of DPPH (100 µM) and ABTS radicals (ABTS (7 mM) and potassium persulfate (2.45 mM)) separately and incubated for 10 min in a dark condition. After incubation, the DPPH and ABTS radicals’ scavenging abilities were determined by measuring the absorbance of 517 nm and 734 nm, respectively. The percentage of radical scavenging activity was determined according to the earlier reports [30].

#### 2.7.3. Cell Viability Assay

The effect of CS, SA, CS-TTO, and SA-CS-TTO on NIH3T3 cells was determined through cell viability assay using a WST assay kit [31]. In brief, the cells were cultured on a T-25 culture flask containing DMEM added with FBS and antibiotic in a CO_2_ incubator at 37 °C. After reaching the cell confluence, 100 µL of cell suspension (1.9 × 10^5^) was seeded to each well in the 96 well plates and kept in the CO_2_ incubator overnight at 37 °C. Then, the cells were incubated with different concentrations of samples containing DMEM for 24 h. Later, the cells were treated with 10 µL of cell viability assay solution (WST) and incubated for 60 min. After incubation, the cell culture plate was measured at 450 nm in the UV-Visible spectrophotometer. The cell viability percentage of each sample was calculated according to the earlier method [31]. The apoptotic effect of the sample on NIH3T3 cells was further evaluated using fluorescent staining (AO/EB, PI, Rh123, and DCFH-DA). In brief, cells were seeded on a 24-well plate, treated with each sample, and incubated with the above-mentioned conditions. After incubation, cells were visualized in a fluorescence-equipped microscope (Olympus, CKX53 culture microscope, Tokyo, Japan).

#### 2.7.4. Wound Healing Assay

The in vitro wound healing/cell proliferation properties of CS, SA, TTO, CS-TTO, and SA-CS-TTO treated NIH3T3 cells were determined through in vitro scratch assay [31,32]. In brief, cells were cultured in 24 well plates containing DMEM, added FBS and antibiotic, and cultured at the above-mentioned condition for 24 h. After confirming the complete confluence of the cell monolayer, the middle of the well plate cells was scratched vertically using a sterile pipette tip. In addition, the cell debris was removed using PBS. Further, the scratched cell monolayer was incubated with DMEM containing each sample and kept in a CO_2_ incubator. The control cells received the DMEM without any samples. Subsequently, the cell proliferation on the scratched area was observed, and images were captured at a different predetermined time interval. The area of cell proliferation was measured using ImageJ software. Further, the percentage of cell proliferation was determined according to an earlier method [31].

### 2.8. Statistical Analysis

All the experiments were repeated three times, and the data in the results presented mean ± standard deviation (SD). The significance of data was analyzed using a one-way analysis of variance (ANOVA). In this study, data were considered statistically significant when the *p* value is <0.05.

## 3. Results

### 3.1. Functional Properties of CS-TTO Nanoemulsion and SA-CS-TTO Microsphere

The functional properties and their changes in the combination were studied through FTIR analysis (Figure 1). CS has two functional groups such as amine (C-2 position) and hydroxyl (C-6 and C-3 position), where chemical modification takes place [33]. CS showed the characteristic medium broad peak at 3291 cm^−1^ attributed to the overlay of O-H and N-H stretching that was due to the hydroxyl and amine groups of CS, respectively [34]. The peak at 2872 cm^−1^ corresponded to the C-H stretching. The band at 1647 cm^−1^ and 1579 cm^−1^ corresponded to the C=O and N-H stretching of the amide I and amide II groups of chitosan, respectively [25,35]. The minor peaks at 1375 cm^−1^ and 1307 cm^−1^ corresponded to the deformation of CH_3_ into CN- stretching, respectively. The broad peak, at 1025 cm^−1^, was attributed to the C-O and C-O-C stretching of CS (Figure 1).

The peak at 894 cm^−1^ and below in the CS was due to the outline C-H bending [36]. TTO showed the medium broad peak at 3483 cm^−1^ corresponding to the O-H stretching in derivatives of a terpene [14]. The sharp peaks at 2960 cm^−1^, 2915 cm^−1^, and 2877 cm^−1^ were attributed to C-H stretching o-cymene in TTO. The peak at 1643 cm^−1^ corresponded to the C=O stretching. The strong, sharp peaks of 1436 cm^−1^ and 1377 cm^−1^ were assigned to the O-H stretching of terpinene 4-ol and other terpene derivatives in the TTO [27]. Then, other medium peaks at 1221 cm^−1^ and 1068 cm^−1^ corresponded to the C-O and CO-O-CO stretching, respectively. Followed by the peaks at 886 cm^−1^ and 798 cm^−1^ corresponding to the C-H bending. The study found that TTO had the several terpene class compound, such as α-pinene, β-pinene, α-terpinene, γ-terpinene, o-cymene, limonene, eucalyptol, terpinolene, terpine 4-ol, and camphene [14]. CS-TTO showed broad peaks at 3267 cm^−1^ due to the N-H stretching of CS. The new peak appeared at 2926 cm^−1^ with 2885 cm^−1^ due to the multiple C-H stretching in the CS-TTO NEMs (Figure 1). In addition, carbonyl (1640 cm^−1^) and amide (1561 cm^−1^) groups shifted, and decreased intensity appeared compared to chitosan owing to the deformation of amide II and amide I. The multiple merged peaks 1464 cm^−1^, 1246 cm^−1^, 1043 cm^−1^, and 841 cm^−1^ were owing to the formation of the CS–TTO complex. SA showed the medium broad characteristic peak of 3248 cm^−1^ assigned to the O-H stretching of the hydroxyl group of alginates. The medium sharp peaks at 1593 cm^−1^ and 1405 cm^−1^ corresponded to the carboxylate ions of asymmetric and symmetric stretching vibration, respectively [37]. The peaks at 1298 cm^−1^, 1024 cm^−1^, and 885 cm^−1^ corresponded to the C-O and C-O-H stretching and C-H bending, respectively. When SA was combined with CS-TTO exhibited the major peaks of 3253 cm^−1^, 1587 cm^−1^, 1410 cm^−1^, and 1022 cm^−1^ (Figure 1). These peaks were highly similar to the alginate with minor shifts while encapsulating the CS-TTO. In SA-CS-TTO, the medium peak that occurred at 3253 cm^−1^ was associated with the formation of hydrogen bonds due to the interaction of amino and hydroxyl groups of CS and SA, respectively. However, the peaks related to the CS-TTO did not appear in CS-TTO-SA due to the strong encapsulation by the SA.

### 3.2. Crystalline Properties CS-TTO and SA-CS-TTO Microsphere

The crystalline structure of the SA-CS-TTO microsphere and its combination material were evaluated through XRD analysis (Figure 2). CS demonstrated a highly stable polymer owing to the coexistence of an amorphous crystalline structure [38]. The results were showed that CS had diffraction peaks at two theta of 9.92° and 19.89°, which due to the hydrogen bond between the hydroxyl and amino groups [39]. Moreover, these peaks of 9.92° and 19.89° in CS corresponded to the crystal plane of 020 and 110, respectively [40]. SA exhibited amorphous crystalline peaked at 13.56° and 21.87°. Similarly, the study supported the characteristic peaks of sodium alginate found at 14° and 22° [41]. The crystalline peaks of CS were weak and shifted when prepared the CS-TTO NEMs which was due to the deformation of the hydrogen bond in the polymer. Similarly, amino–thiol functionalization on chitosan decreased the amorphous crystalline structure compared to chitosan [42]. Furthermore, a combination of CS-TTO with SA showed weaker amorphous crystalline peaks at 7.1° and 14.40°. Those peaks were completely shifted compared to the unaltered CS and SA. In addition, the previous study indicated that CS–SA hydrogel showed no noticeable crystal peaks [43].

### 3.3. CS-TTO NEMs Size Distribution by NTA

NTA is an analytical method to analyze the nanoparticle’s size distribution in liquid suspension through the Brownian motion and light scattering techniques [44]. In addition to that, it assists in measuring the concentration of particles in any liquid suspension/NEMs. Hence, the particle size distribution and count of the CS-TTO NEMs were determined under NTA (Figure 3A). The results (size (nm) vs. concentration (particles/mL)) showed the CS-TTO NEMs had an average particle size of 89.5 nm (Figure 3A(a,b)). In addition, CS-TTO NEMs exhibited the concentration of particles/frame was 57.2 ± 2.2, and the concentration of the particles/mL was 3.76 × 10^8^. Further, the measured median diameter (D50) of the particles was 74.2 nm (Figure 3A(a)). Similar to the finding, amphotericin B-loaded NEMs found an average particle size of 135 nm in the NTA [45]. In addition, the study supported that the EO-based NEMs from the leaves of *Cymbopogon densiflorus* showed the particle concentration/mL was 10^12^ with a mean size of 71 nm [46]. Further, the study was performed to identify the NEMs concentration and size and support the other instrumental results such as DLS and TEM [47].

### 3.4. SEM Analysis of CS-TT-SA Microsphere

The morphology of SA-encapsulated CS-TTO NEMs was observed under SEM (Figure 3B). The SEM micrograph displayed that the freeze-dried SA-CS-TTO microsphere had a smooth surface with distinct spherical and ovoid morphology (Figure 3B(a,b)). The particle size distribution of the SA-CS-TTO microsphere was found to be from 0.5 to 1.0 µm, and the average size of 0.76 ± 0.10 µm (Appendix A). Similarly, TTO loaded quaternary ammonium salt of CS and SA microcapsules spherical particles with the size of 1.91 to 13.18 µm [27]. These particles’ size was larger than SA-CS-TTO microsphere which might be the difference in the synthesis method. The particle size of the microsphere increased when loading the TTO into the prepared beads, while the size decreased when prepared CS-TTO NEMs encapsulated the SA microsphere. Another study evidenced that porous and rough-surfaced spherical shape particles were obtained while physical crosslinking of SA, carboxymethyl chitosan and collagen [48]. Furthermore, we found that the microsphere size was increased depending on the concentration of sodium alginate. In addition, ultra-sonication assists in the dispersion of the microsphere from the aggregation.

### 3.5. Water Absorption, In Vitro Degradation, and Thermal Stability

The water absorption/swelling properties are important characteristics of wound dressing material. The swelling properties aid in absorbing the excess amount of wound exudates from the wounded area and provide oxygen exchange, and maintain the wet environment, which exerts cell proliferation. The results indicated that the combination of CS, TTO, and SA increased the water absorption ability depending on the incubation time (Figure 4a). The percentage of water absorption was noted at 710 ± 87% for the first hour and increased to 1800 ± 178% for the third hour (*p* < 0.001). Following that, water absorption was not significantly increased with further incubation time (*p* < 0.05). In addition, the SA-CS-TTO exhibited the maximum water absorption percentage of 2000 ± 170% at the fifth hour. The water absorption properties of SA-CS-TTO highly depend on the hydroxyl and amino functional group of materials [49]. Moreover, the previous study found that the combination of carboxymethyl CS-SA improved the swelling ability of the microsphere, while it decreased when cross-linked with Ca [50]. The in vitro degradation of SA-CS-TTO was evaluated in a lysozyme-incorporated phosphate buffer medium (pH 7.4) until 28 days (Figure 4b). The results showed that the percentage of SA-CS-TTO degradation increased depending on the duration of incubation. The degradation of SA-CS-TTO was found to be 3.8 ± 0.6% and 14.3 ± 1.1% for the 7th and 28th days, respectively. However, the degradation rate of SA-CS-TTO was decreased due to the combinational effect of CS, SA, and TTO. The cross-linking ability of co-polymers is restricted to enzymatic degradation [50]. Similarly, the chitosan and alginate hydrogel microsphere maintained stability in terms of initial mass until 21 days [51].

The thermal stability of SA-CS-TTO was determined through TGA (Figure 4c). The results indicated that the SA-CS-TTO had four stages of thermal degradation. The first stage of maximum thermal degradation in terms of weight loss of 13.84% was found at 141 °C due to the loss of surface and inner particles moisture and depolymerization (breakage of mannuronic acid and glucuronic acid unit) of SA [52]. The major weight loss (41.88%) occurred in the second stage with the temperature at 142–384 °C owing to the degradation of polymer structure (cleavage of glycosidic bonds and saccharide rings) of CS and SA. In that third stage, the minimum amount of weight loss (7.984%) was found at 644 °C. Finally, a medium amount of weight loss (10.82%) was found in the fourth stage at a temperature of 800 °C. Interestingly, 25.47% of residues from SA-CS-TTO were found after 800 °C of thermal degradation. Similarly, CS/SA freeze-dried aerogel exhibited major thermal degradation at 260 °C [53]. These results indicated that the combination of CS and SA improves the thermal stability combined with TTO. Further, the polymer combination evidenced that it improves the stability of TTO even at a high temperature.

### 3.6. TTO Encapsulation, Loading, and In Vitro Release

The percentage of TTO encapsulation and loading efficiency on SA-CS-TTO were 71.1 ± 3.4% and 14.47 ± 0.94%, respectively. Further, the release rate of TTO from the SA-CS-TTO was evaluated through in vitro release analysis (Figure 4d). The results showed that TTO released from the SA-CS-TTO microsphere depends on the temperature and in a sustained manner. The release rate of TTO from SA-CS-TTO at 25 °C was lower than 37 °C (*p* < 0.05) due to the SA, which degrades at higher temperatures. The release pattern of TTO from SA-CS-TTO at 120 h was found at 51.5 ± 0.5% and 66.4 ± 2.8% for 25 °C and 37 °C, respectively. In addition, the release pattern of TTO from the polymeric system highly depends on the physical properties of complex material. The study supported that the release of TTO is influenced by the monomer ratio and cross-linking agent [54]. In addition, the cumulative release of TTO from TTO/liposomes-impregnated chitosan nanofiber was found to be higher at 25 °C compared to 4 °C [55]. Similarly, the release rate of TTO increased while increasing the temperature (4 °C, 25 °C, 37 °C) and humidity (24%, 35%, and 50%) [27].

### 3.7. Antibacterial Activity

Antimicrobials play a major role in accelerating wound healing through the downregulation of the proteolytic enzyme by reduction of microbial colonization [56]. However, often the administration of particular types of antibiotics leads to antibiotic resistance and causes a potential impact on chronic wound healing [57]. To eliminate these drawbacks, combinational antibiotics are being used with anti-inflammatory agents [58,59]. In this regard, natural antimicrobials (biomolecules, peptides, lipids, and polysaccharides), specifically essential oil, contain various derivatives that exert antimicrobial as well as anti-inflammatory activity by escaping antimicrobial resistance and down-regulation of the inflammatory cytokines [60,61]. Hence, the antibacterial potential of prepared CS-TTO and SA-CS-TTO, along with TTO, was evaluated against various bacterial pathogens using different antibacterial assays. TTO exhibited strong bacterial inhibitory activity against a wide range of bacterial species. Particularly, our previous study found that TTO showed antibacterial activity, including foodborne bacterial pathogens *B. cereus*, *S. aureus*, *E. coli*, *S. enterica*, and *L. monocytogenes* [8]. CS-TTO significantly inhibited the growth of bacterial pathogens, which depended on the concentration (Appendix A). Importantly, the concentration of TTO in CS-TTO NEMs was very low compared to TTO. These results indicated that the use of CS-based TTO NEMs increases the antibacterial properties with minimum concentration, and it could avoid the potential toxicity by direct use of TTO [15]. The CS-TTO NEMs (100 µg/mL) showed a zone of inhibition of 16.2 ± 1.8 mm, 15.4 ± 1.2 mm, 12.3 ± 1.5 mm, 13.1 ± 1.4 mm, 13.6 ± 1.1 mm for *B. cereus*, *S. aureus*, *E. coli*, and *S. enterica*, respectively (Appendix A). In addition, to evaluate the MIC_50_ of different concentrations of CS, SA, TTO, CS-TTO, and SA-CS-TTO, they were incubated with bacterial pathogens. The MIC of CS against *B. cereus*, *S. aureus*, and *E. coli* was similar to SA, while CS showed less MIC than the SA against *S. enterica* (*p* < 0.05). Further, MIC_50_ of CS-TTO was found at 62.5 µg/mL, 62.5 µg/mL, 62.5 µg/mL, and 31.25–62.5 µg/mL for *B. cereus*, *S. aureus*, *E. coli*, and *S. enterica*, respectively (Table 1). These CS-TTO NEMs showed less MIC concentration compared to TTO alone. However, the MIC_50_ of SA-CS-TTO was similar to TTO (*p* < 0.05) due to the sustained release.

Further, the antibacterial activity of CS, SA, TTO, CS-TTO, and SA-CS-TTO against *S. aureus* and *E. coli* were determined in CLSM using Syto-9/PI fluorescent staining assay (Figure 5 and Figure 6). The results indicated that SA did not show antibacterial activity against both bacteria compared to the control, while CS exhibited a few bacterial deaths (Figure 5 and Figure 6). TTO, CS-TTO, and SA-CS-TTO caused notable bacterial cell death irrespective of the bacterial species. Among them, CS-TTO NEMs and SA-CS-TTO caused higher cell death than TTO, which might be the reason for releasing the TTO from the CS-TTO NEMs attached to the surface of bacterial pathogens. The earlier work evidenced that TTO-loaded SA/quaternary ammonium salt of CS significantly inhibited both *S. aureus* and *E. coli* but showed higher activity in *S. aureus* due to the breakable cell membrane [27]. Similarly, carboxymethyl-CS-genipin encapsulated TTO showed higher inhibitory activity against *S. aureus* (99%) than *E. coli* (90%) [62]. TTO incorporated CS film with significant antibacterial and antifungal activity against *L. monocytogenes* and *Penicillium italicum*, respectively [63]. Hence, these results concluded that TTO exhibits significant antibacterial activity, and TTO encapsulated microsphere ensured the antibacterial efficacy.

### 3.8. Antioxidant Activity

Antioxidant molecules are essential for attenuating oxidative stress during the inflammatory response and stimulating cell proliferation. Therefore, CS, SA, TTO, CS-TTO, and SA-CS-TTO determined DPPH and ABTS radical scavenging activity to understand the antioxidant properties (Figure 7). The results indicated that all the base materials showed DPPH radical scavenging activity depending on the concentration (Figure 7a). Among them, TTO and CS-TTO showed significant DPPH radical scavenging activity compared to SA-CS-TTO, which was due to the sustained release of TTO by the SA encapsulation. The percentage of DPPH radical scavenging activity of SA-CS-TTO was close to CS (Figure 7a). TTO reported their potent antioxidant activity owing to the terpine derivatives in the composition (α-terpinolene, α-terpinene, and γ-terpinene) [64]. In addition, the 100 µL/mL of methanol-dissolved TTO showed that the DPPH radical scavenging activity was similar to the synthetic antioxidant molecules BHT (butylated hydroxytoluene) [64]. The IC_50_ concentration of SA was not found within the tested concentration (50 µg/mL) against the DPPH. Among them, the IC_50_ concentration of CS-TTO NEMs and TTO in DPPH radical scavenging was 24 µg/mL and 26 µg/mL, respectively. However, the ABTS radical scavenging ability of the tested material differed from the pattern of DPPH radical scavenging. Interestingly, the lower concentration of TTO (3.12 µL/mL) had 74.96 ± 2.60% of ABTS radical scavenging ability compared to CS-TTO NEMs (46.41 ± 1.56%) and SA-CS-TTO (48.40 ± 1.79%). In addition, ABTS radical scavenging activity of CS and SA was found at 29.91 ± 3.87% and 44.39 ± 1.72%, respectively (Figure 7b). Although the ABTS radical scavenging ability of TTO was higher in the lower concentration, it did not significantly increase depending on the concentration (*p* < 0.05); TTO showed the maximum ABTS radical scavenging activity of 95.62 ± 1.09%. On the other hand, TTO-loaded CS-TTO NEMs and SA-CS-TTO increased ABTS radical scavenging activity while increasing the concentration (Figure 7b). The CS-TTO increased the radical scavenging activity through the amine and phenolic group of CS and TTO strongly binding with hydroxyl (OH) and DPPH radicals [62]. The study supported that clove essential oil (CEO) loaded chitosan nanoparticle increases the DPPH radical scavenging activity more than the bare CEO [65]. These results indicated that polymer encapsulation limits the complete release of TTO. Hence, the antioxidant activity of SA-CS-TTO was not higher than that of bare TTO, but it could inhibit the free radical for a longer time.

### 3.9. Cell Viability

The cell viability effect of CS, SA, TTO, CS-TTO, and SA-CS-TTO treated NIH3T3 cells are shown in Figure 8. The results showed that the treatment of polymer (SA and CS) and TTO-loaded polymer with their increasing concentration (3.12–100 µg/mL) did not inhibit the cell viability of NIH3T3 cells. However, the increasing concentration of TTO alone significantly caused cell toxicity while not observed in the TTO and polymer combination due to the antioxidant and anti-inflammatory properties of CS and SA [66,67]. Further, CS-TTO and SA-CS-TTO did not cause the NIH3T3 cell viability reduction even at a higher concentration of 100 µg/mL. This phenomenon might be because of the sustained release of TTO from the SA-CS-TTO improved cell proliferation due to the antioxidant properties.

### 3.10. In Vitro Wound Healing Assay

The in vitro wound healing ability of CS, SA, CS-TTO, and SA-CS-TTO were determined on NIH3T3 cells (Figure 9). In the cell viability assay, we found that the TTO alone treatment caused cytotoxicity in NIH3T3 cells. Hence, in vitro wound healing assay TTO was excluded and was not compared with other materials. The ability of cell proliferation after scratching the cells with the treatment of different materials was determined at different intervals (6 h, 18 h, 24 h, and 36 h) (Figure 9a). In the first interval (6th h), SA alone exhibited higher cell coverage (8.48 ± 1.34%) compared to the control and other treatments (*p* < 0.01). However, CS alone showed more cell coverage of 57.73 ± 2.63% and 64.13 ± 1.95% for the second (18th h) and third intervals (24th h), respectively. Interestingly, in the fourth interval (36th h), the maximum cell coverage percentage was observed in the following order, which was 98.05 ± 1.2 1% for SA-CS-TTO, 95.97 ± 0.31% for CS, 93.56 ± 1.52% for CS-TTO, 88.93 ± 3.71% for SA, and 86.11 ± 2.59% for control cells (*p* < 0.01) (Figure 9b). These results indicated that the longer incubation time for SA-CS-TTO might increase the release/dispersion of TTO, CS, and SA in the medium, which induced higher cell proliferation of NIH3T3 cells. Similarly, the study evidenced that subdermal implantation of CS/polyvinyl Alcohol/TTO composite film decreased the inflammatory phase and improved biocompatibility [12]. In addition, carboxymethyl-CS-genipin encapsulated TTO induced significant wound closure in rat models due to the antibacterial, antioxidant, and anti-inflammatory properties of TTO and the degradation of polymers (SA and CS) [62,66]. Hence, these results concluded that the combination of SA-CS-TTO could improve wound healing ability owing to their antioxidant, anti-inflammatory, and antibacterial properties.

## 4. Conclusions

The TTO-loaded CS NEMs were prepared and encapsulated by the SA successfully for the antioxidant and antibacterial wound dressing. The NTA evidenced the formation of CS-TTO NEMs with an average size of 89.5 nm. Further, SA encapsulated CS-TTO microsphere had spherical particles with an average size of 0.76 ± 0.10 µm. The amorphous crystalline properties of CS and SA diminished when combined with TTO. The CS-TTO-SA exhibited moderate water swelling and a low degradation rate in lysozyme incorporated phosphate buffer medium (pH 7.4). Further, these microspheres released the TTO in a sustained manner in PBS at 25 °C. The TTO and TTO incorporated CS-SA showed radical scavenging activity against DPPH and ABTS radical and antibacterial activity against Gram-positive and Gram-negative pathogens. Furthermore, SA-CS-TTO microsphere improved cell proliferation and exhibited significant in vitro wound healing activity on NIH3T3 cells. These results indicated that the prepared combination of CS, TTO, and SA could be used for the potential antimicrobial wound dressing for in vivo experiments.

## Figures and Tables

**Figure 1 nutrients-15-01319-f001:**
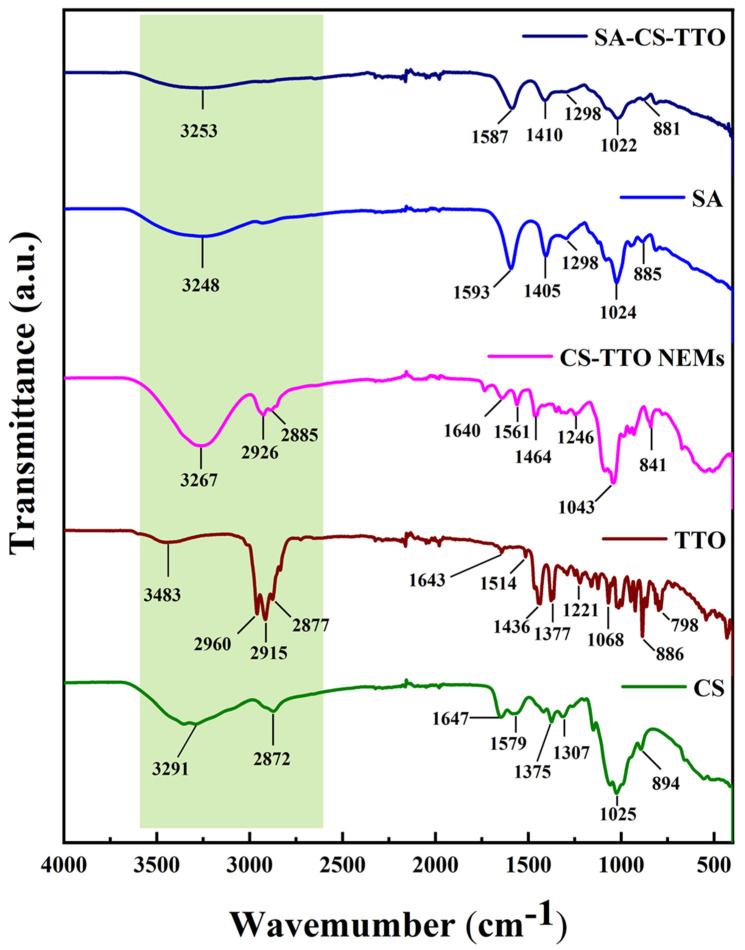
FTIR spectrum of chitosan (CS), tea tree oil (TTO), CS-TTO nanoemulsion (CS-TTO NEMs), sodium alginate (SA), and SA-CS-TTO microsphere.

**Figure 2 nutrients-15-01319-f002:**
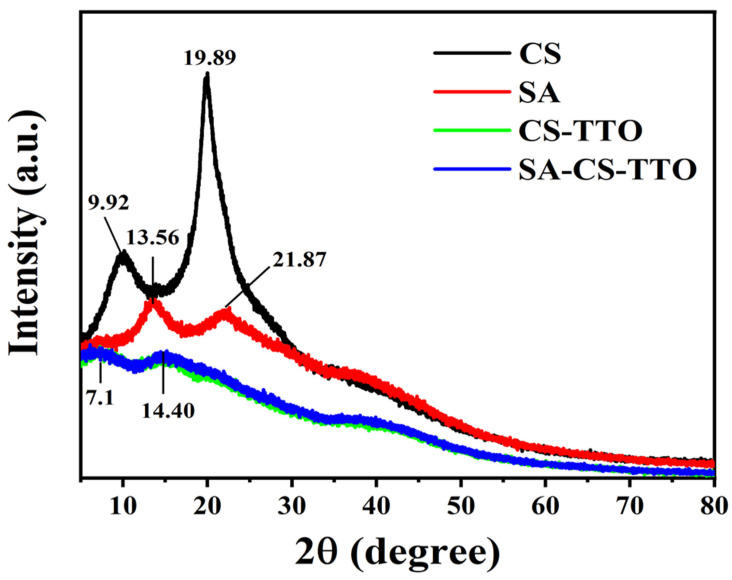
XRD spectrum of chitosan (CS), sodium alginate (SA), CS emulsified tea tree oil (TTO), SA encapsulated CS-TTO microsphere.

**Figure 3 nutrients-15-01319-f003:**
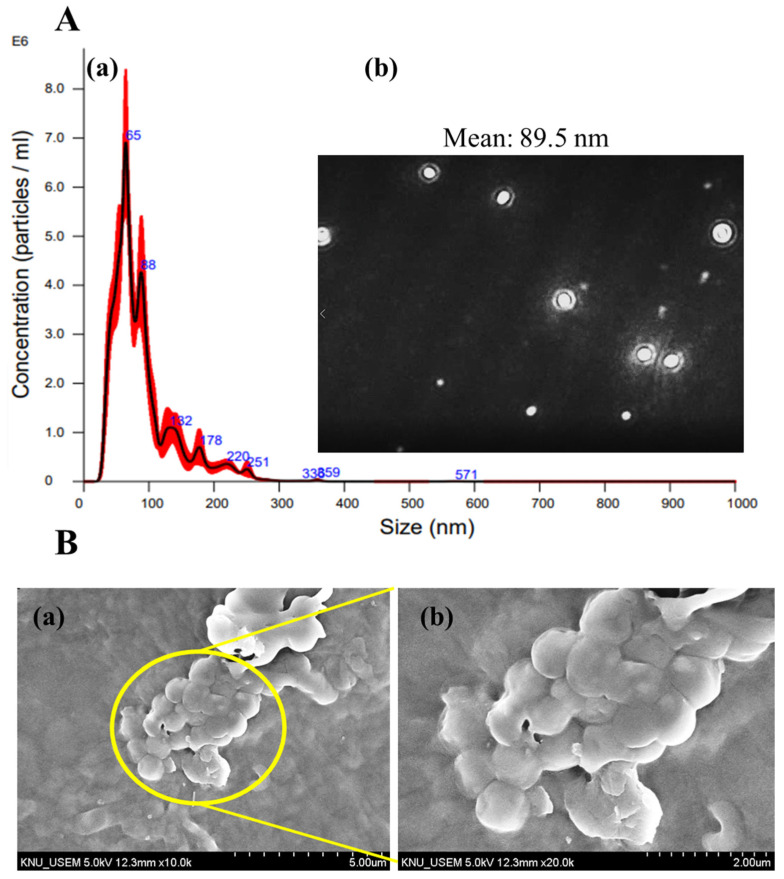
Nanoparticle tracking analysis (NTA)—(**A**). The spectrum and image of CS-TTO NEMs—(**a**,**b**); SEM micrograph of sodium alginate microencapsulated CS-TTO microsphere—(**B**). The spherical particle of SA-CS-TTO microsphere—(**a**); and its higher magnification—(**b**). Chitosan (CS); sodium alginate (SA); tea tree oil (TTO).

**Figure 4 nutrients-15-01319-f004:**
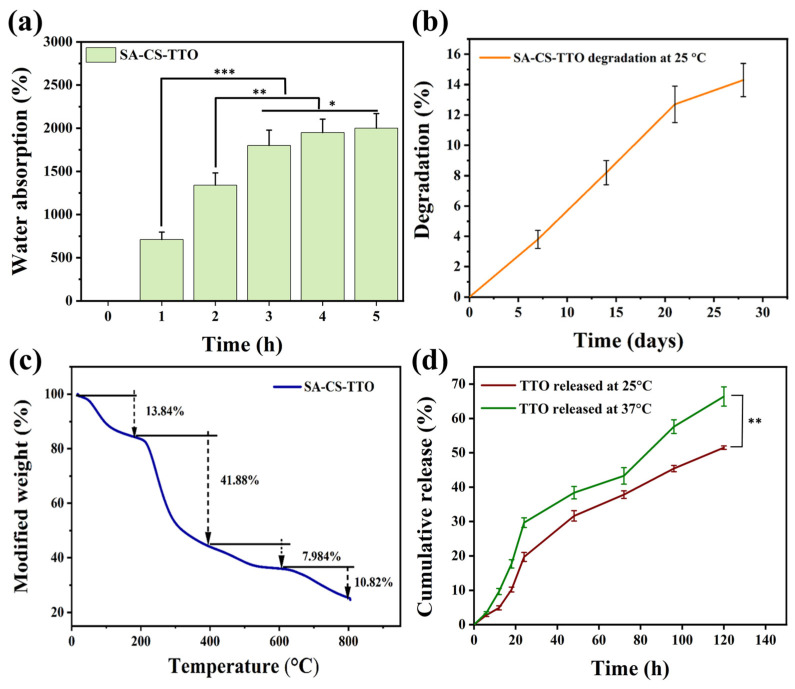
Characterization of SA-CS-TTO microsphere. Water absorption—(**a**); in vitro degradation—(**b**); thermogravimetric analysis—(**c**); TTO release at 25 °C and 37 °C—(**d**). Chitosan (CS); sodium alginate (SA); tea tree oil (TTO). Statistical significance considered as * *p* < 0.05, ** *p* < 0.01, *** *p* < 0.001.

**Figure 5 nutrients-15-01319-f005:**
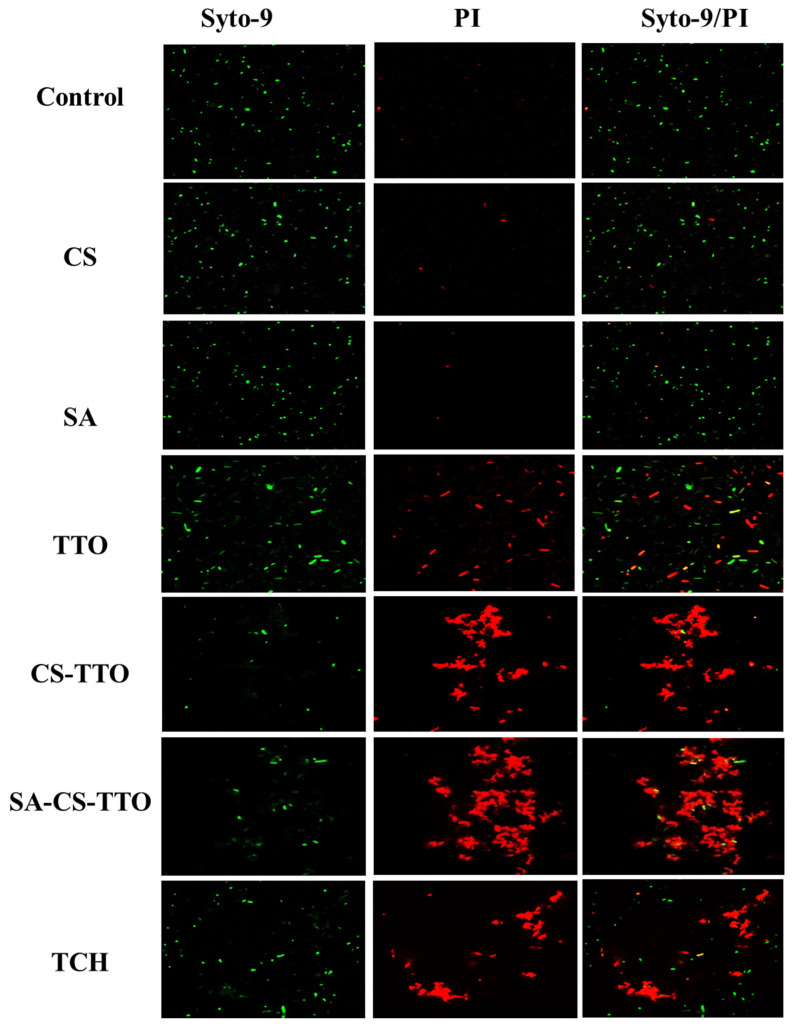
Antibacterial activity of microsphere against *S. aureus* observed under CLSM analysis. The effect of the SA-CS-TTO microsphere and its composition was evaluated using live and dead assay staining (Syto-9/PI). Chitosan (CS); sodium alginate (SA); tea tree oil (TTO).

**Figure 6 nutrients-15-01319-f006:**
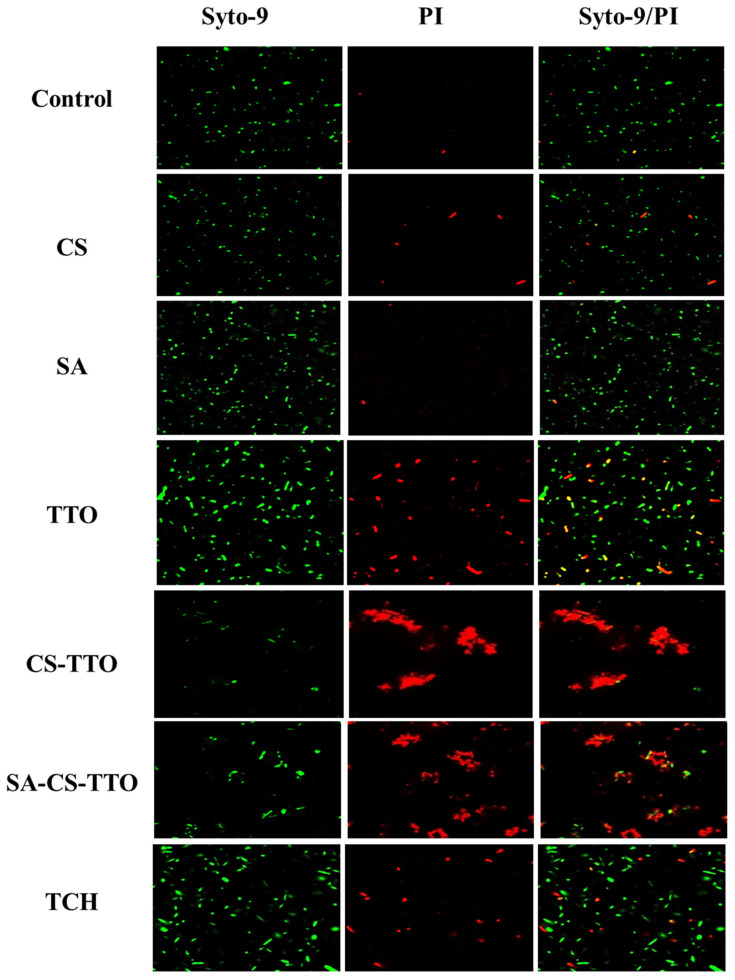
Antibacterial activity of microsphere against *E. coli* observed under CLSM analysis. The effect of the SA-CS-TTO microsphere and its composition was evaluated using live and dead assay staining (Syto-9/PI). Chitosan (CS); sodium alginate (SA); tea tree oil (TTO).

**Figure 7 nutrients-15-01319-f007:**
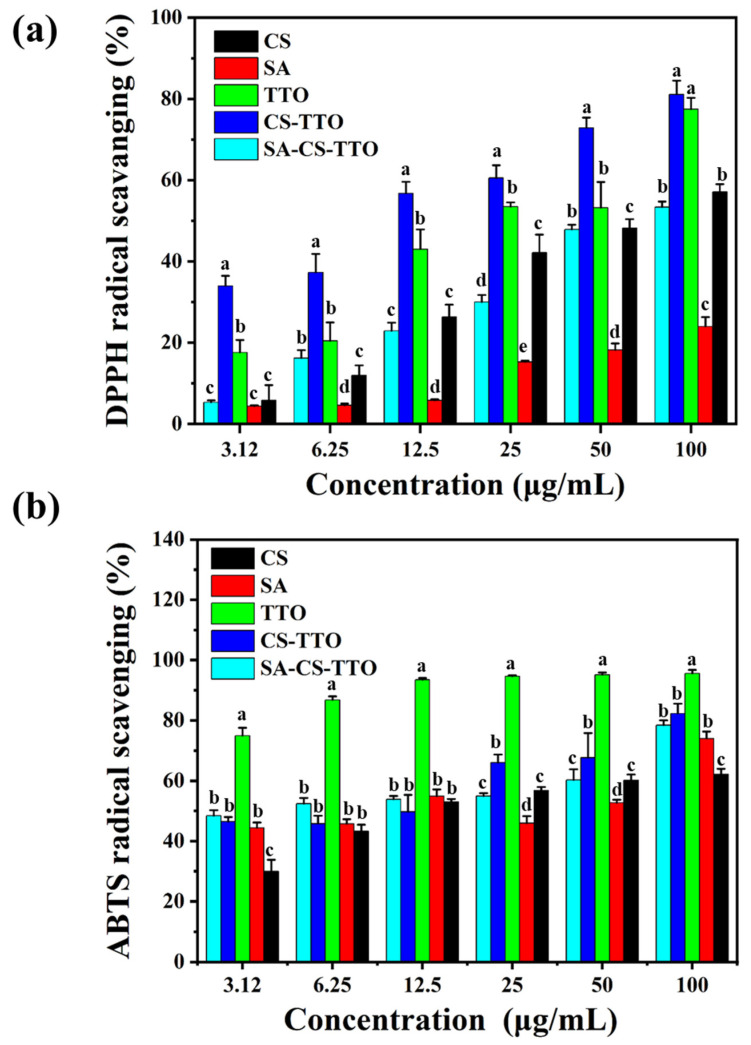
Antioxidant activity of SA-CS-TTO microsphere and its composition. DPPH radical scavenging assay—(**a**); ABTS radical scavenging assay—(**b**). Chitosan (CS); sodium alginate (SA); tea tree oil (TTO). The letters in each column significantly differ (*p* < 0.05) among the samples.

**Figure 8 nutrients-15-01319-f008:**
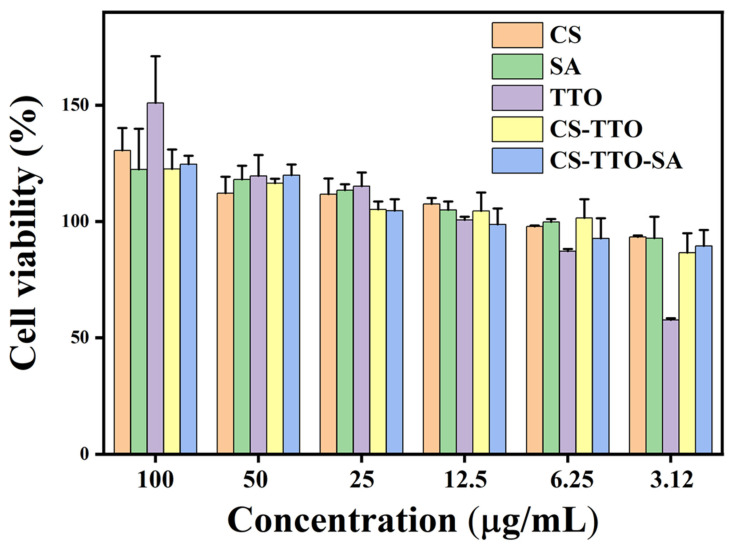
The cell viability of SA-CS-TTO microsphere and base materials in NIH3T3 cells. Chitosan (CS); sodium alginate (SA); tea tree oil (TTO).

**Figure 9 nutrients-15-01319-f009:**
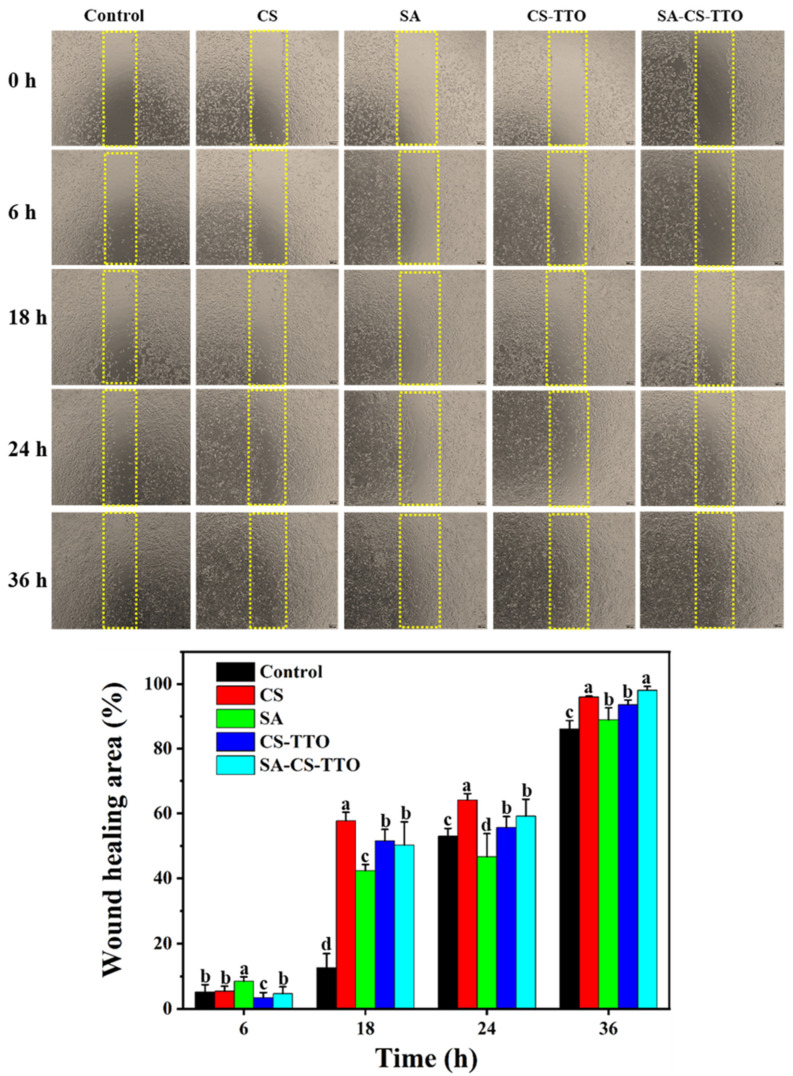
In vitro wound healing analysis. Control cells; CS treated cells; SA treated cells; CS-TTO treated cells; and CS-TTO-SA treated cells. Chitosan (CS); sodium alginate (SA); tea tree oil (TTO). The letters in each column significantly differ (*p* < 0.01) among the samples.

**Table 1 nutrients-15-01319-t001:** Minimum inhibitory concentration (MIC) of various samples against bacterial pathogens. Values followed by the superscript letters in the column significantly differ (*p* < 0.05) among the samples.

Sample	*B. cereus*	*S. aureus*	*E. coli*	*S. enterica*
	MIC (µg/mL)			
CS	250 ^b^	250 ^c^	>250 ^c^	125 ^b^
SA	250 ^b^	>250 ^c^	>250 ^c^	>250 ^c^
TTO	62.5 ^a^	125 ^b^	125 ^b^	125 ^c^
CS-TTO	62.5 ^a^	62.5 ^a^	62.5 ^a^	31.25–62.5 ^a^
SA-CS-TTO	62.5 ^a^	125 ^b^	125 ^b^	125 ^b^

## Data Availability

The data presented in this study are available on request from the corresponding author.

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
