# Peer review of "Enhancing the Antioxidant, Antibacterial, and Wound Healing Effects of Melaleuca alternifolia Oil by Microencapsulating It in Chitosan-Sodium Alginate Microspheres"

_nutrients, 2023, doi:10.3390/nu15061319_

Round 1

Reviewer 1 Report

The article is well written and the scientific methodologies are adequate. In particular, the characterization techniques as well as the reported in vitro applications are of particular interest.  In particular:

- acronyms need to be spelled out better

- in figure 1 it is difficult for me consider a peak the signal at 3253 for the SA-CS-TTO. Authros should better explain the extrapolated conclusion relatd to this value

Author Response

The article is well written and the scientific methodologies are adequate. In particular, the characterization techniques as well as the reported in vitro applications are of particular interest.  In particular:

- acronyms need to be spelled out better

Response: Thanks for your keen observation and valuable suggestions. The revised manuscript updated according to the reviewer suggestions.

- in figure 1 it is difficult for me consider a peak the signal at 3253 for the SA-CS-TTO. Authors should better explain the extrapolated conclusion related to this value.

Response: Thank you for your query. FTIR section updated with relevant conclusion specified with the peak at 3253 for the SA-CS-TTO in the revised manuscript.

Thank you very much for your valuable comments on improving our articles in a better way. We believe that we have answered all the queries asked by the reviewers and made all the changes in the manuscript as per the Editor and reviewer’s suggestions.

Thank you so much to the reviewers and the Editor

Prof. M. H. Wang

Reviewer 2 Report

The manuscript from Sathiyaseelan describes the formulation of chitosan-sodium alginate microspheres containing Melaleuca alternifolia oil in nanoemulsion. The formulation showed enhanced in vitro antioxidant, antibacterial, and wound healing effects.

The manuscript has several points that need to be accepted before the final decision could be taken.

Abstract 

- Provide the species name at the first citation 

Introduction

- The authors must rewrite some sentences in the second paragraph. They are indicated in the attached pdf file.

Methods 

- Please attempt to the use of italic for Latin names. Do this throughout the manuscript.

- The formula in section 2.5 was not provided.

- Please provide the codes for each strain used. Do this throughout the manuscript.

- Please provide the appropriate reference for section 2.7.2 which have the instructions for the calculation of antioxidant activities.

Results

- The statistical analysis is not indicated in the figures. The results also is not compared in the text using the statistical analysis.

- The table of antimicrobial activity is not provided.

- The titles of figures 5, 6 and 8 need to be revised.

Discussion

The discussion section is missed. The results need to be discussed in a comprehensive way.

Other comments are provided in the attached pdf file.

Author Response

Reviewer – 2

he manuscript from Sathiyaseelan describes the formulation of chitosan-sodium alginate microspheres containing Melaleuca alternifolia oil in nanoemulsion. The formulation showed enhanced in vitro antioxidant, antibacterial, and wound healing effects. The manuscript has several points that need to be accepted before the final decision could be taken.

Abstract

- Provide the species name at the first citation

Response: Thanks for your valuable comments. The species name has been provided in the revised version.

Introduction

- The authors must rewrite some sentences in the second paragraph. They are indicated in the attached pdf file.

Response: Authors really appreciate the reviewer for the valuable suggestion and corrections in the pdf file. We have corrected according the reviewer suggestion in the revised manuscript.

Methods

- Please attempt to the use of italic for Latin names. Do this throughout the manuscript.

Response: Thanks for your valuable comments. Actually, we used italics for Latin names but all changed while pasting in the journal template. However, we are sorry for the mistakes and unnoticed submission. These mistakes in the whole manuscript has been corrected.

- The formula in section 2.5 was not provided.

Response: We apologize for this mistake. The formula for in vitro degradation of microsphere has been added in the revised version. We appreciate your valuable comments.

- Please provide the codes for each strain used. Do this throughout the manuscript.

Response: The bacterial strains were obtained from the American Type Culture Collection. The ATTC accession number has been provided for each bacterial strain.

- Please provide the appropriate reference for section 2.7.2 which have the instructions for the calculation of antioxidant activities.

Response: The appropriate citation has been provided for section 2.7.2. Thank your keen observation.

Results

- The statistical analysis is not indicated in the figures. The results also is not compared in the text using the statistical analysis.

Response:

- The table of antimicrobial activity is not provided.

Response: Thank you for your keen observation. The zone of inhibition table has been provided for antimicrobial activity in the revised version.

- The titles of figures 5, 6 and 8 need to be revised.

Response: Thank you for your suggestions. We have corrected the title of figures 5, 6 and 8 in the revised version.

Discussion

The discussion section is missed. The results need to be discussed in a comprehensive way.

Response: Thanks for your suggestions. To highlight and compare the results, the discussion section was merged with results parts and presented as “Results and discussion”. Moreover, we have updated the section in a comprehensive way according to the reviewer suggestion.

Other comments are provided in the attached pdf file.

Response: We truly appreciate the reviewer comments and the time spent for reviewing this manuscript.

Thank you very much for your valuable comments on improving our articles in a better way. We believe that we have answered all the queries asked by the reviewers and made all the changes in the manuscript as per the Editor and reviewer’s suggestions.

Thank you so much to the reviewers and the Editor

Prof. M. H. Wang

Round 2

Reviewer 2 Report

The authors have addressed the questions.